# Hard Coating Materials Based on Photo-Reactive Silsesquioxane for Flexible Application: Improvement of Flexible and Hardness Properties by High Molecular Weight

**DOI:** 10.3390/polym13101564

**Published:** 2021-05-13

**Authors:** Jong Tae Leem, Woong Cheol Seok, Ji Beom Yoo, Sangkug Lee, Ho Jun Song

**Affiliations:** 1Green and Sustainable Materials R&D Department/Research Institute of Clean Manufacturing System, Korea Institute of Industrial Technology, 89 Yangdaegiro-gil, Ipjang-myeon, Seobuk-gu, Cheonan-si 331-822, Chungcheongnam-do, Korea; dla700@kitech.re.kr (J.T.L.); vmfosel07@kitech.re.kr (W.C.S.); skdlee@kitech.re.kr (S.L.); 2School of Advanced Materials Science and Engineering, Sungkyunkwan University, Suwon 16419, Gyeonggi-do, Korea; jbyoo@skku.edu

**Keywords:** flexible hard coating, sol–gel reaction, polyhedral oligomeric silsesquioxane, ladder-like polysilsesquioxane

## Abstract

EPOSS of polyhedral oligomeric silsesquioxanes (POSS) mixture structure and LPSQ of ladder-like polysilsesquioxane (LPSQ) structure were synthesized via sol–gel reaction. EPSQ had a high molecular weight due to polycondensation by potassium carbonate. The EPSQ film showed uniform surface morphology due to regular double-stranded structure. In contrast, the EPOSS-coated film showed nonuniform surface morphology due to strong aggregation. Due to the aggregation, the EPOSS film had shorter d-spacing (d1) than the EPSQ film in XRD analysis. In pencil hardness and nanoindentation analysis, EPSQ film showed higher hardness than the EPOSS film due to regular double-stranded structure. In addition, in the in-folding (r = 0.5 mm) and out-folding (r = 5 mm) tests, the EPSQ film did not crack unlike the EPOSS coated film.

## 1. Introduction

Recently, flexible displays, such as curved, foldable, bendable, rollable, and stretchable displays, have attracted attention [1,2,3,4]. Therefore, there have been many studies investigating films [5], adhesives [6], and hard coatings [7] for flexible displays. Among these options, a hard coating is a top layer that not only protects the underlying layers or substrate from chemical damage and UV decomposition but also provides protection from scratches [8]. Generally, hard coatings for displays require various characteristics, such as high transmittance, thermal stability, and hardness [7,9]. On the other hand, hard coatings for flexible displays require flexibility due to various stress conditions, including tensile, compression, and bending [10,11,12].

To achieve these characteristics, organic–inorganic hybrid coating materials that combine the rigid properties of inorganic materials and the flexibility of organic materials have been actively developed [13]. Organic–inorganic hybrid materials are widely used in the coating field because they have high-performance properties that combine the advantages and disadvantages of organic and inorganic materials [14,15]. Silicone is a representative organic–inorganic hybrid material [16]. Silicone compounds, known as silicones or siloxanes, were developed in the early 20th century [17]. Silicon is the most common element in the earth’s crust and contains four electrons in its outermost shell, similar to carbon [18]. However, silicon compounds have better mechanical and thermal properties than organic materials composed of carbon [17,19]. In addition, the Si–O–Si bonding angle is 140 to 180°, which is higher than the C–O–C ether bond angle of 110°; thus, it has high flexibility [20]. Because of these advantages, silicone compounds are widely used in high-performance materials that require high thermal stability and flexibility. Silicone compounds are based on bifunctional and trifunctional silanes for their high-performance properties. Bifunctional silanes are used in the synthesis of linear and cyclic oligosiloxanes and polysiloxanes, while trifunctional silanes are used in the synthesis of oligosiloxanes, polysiloxanes, and polysilsesquioxanes [21,22].

Silsesquioxane has the chemical formula RSiO_1.5_ (R is an organic group or H) [23]. Silsesquioxanes are easier to control than silica, and they are attracting attention because they can improve various properties compared to silica owing to their organic functional groups [24]. In particular, silsesquioxane is highly miscible with a polymer matrix due to its organic functional groups, which can increase solubility in organic solvents [25,26]. Representative silsesquioxane structures are polyhedral oligomeric silsesquioxane (POSS) and ladder-like polysilsesquioxane (LPSQ) [9]. POSS and LPSQ have a chemical structure of (RSiO_1.5_)n consisting of strong Si–O–Si bonding [23,27]. POSS is an oligomer with a chemical structure (RSiO_1.5_)n (N = 6, 8, 10, 12) [28,29]. The POSS structure shows high thermal stability owing to its three-dimensional cage structure [30,31]. On the other hand, the LPSQ structure has a long repeating unit of Si–O–Si and a high molecular weight [32]. In addition, LPSQ has improved optical and thermal stability and mechanical properties owing to its unique double-stranded siloxane structure [32,33,34]. In particular, the double-stranded siloxane structure has high rigidity due to the Si–O internal rotation hindrance compared to that of a single chain [31]. POSS and LPSQ are used in various fields, such as separator, medicine, catalysis, sensor, and coating, due to their unique structure and various advantages [27,28,29,31].

These silicon compounds are generally obtained through sol–gel reactions. Sol–gel reactions were developed in the early 1930s, and most sol–gel reactions proceed under mild conditions [35]. The sol–gel reaction includes the following steps: (a) hydrolysis of the alkoxy group, (b) condensation of two silanol groups, and (c) condensation of the silanol and alkoxy groups. The Si–O–Si bonds of the silicon compound are formed through the sol–gel reaction [36]. In the sol–gel reaction, the hydrolysis reaction is affected by the alkyl group, solvent, temperature, and catalyst [37,38]. For example, with a base catalyst, the condensation rate is higher than the hydrolysis rate. On the other hand, an acid catalyst synthesizes a low-fractal-dimensional structure with many silanol groups on the silica surface due to fast hydrolysis [37]. In general, acid-catalyzed reactions are used to synthesize linear polymers and branched polysiloxanes, and base and heating reactions are widely used to synthesize cubes and ladder-like silsesquioxanes (LPSQs) [21]. Recently, silsesquioxane structure based on 2-(3,4-epoxycyclohexyl)ethyltrimethoxysilane (ECTMS), which has high thermal stability and good mechanical properties, has been widely studied [39,40,41,42].

In the study by Hwang et al., Co-LPSQs synthesized using ECTMS and trimethoxyphenylsilane (PTMS) showed a 4–9H pencil hardness at a load of 7.5 N (coating thickness of 20 μm) [39]. In addition, a siloxane hybrid structure was synthesized using ECTMS by Bae et al. that showed a 9H pencil hardness at a load of 9.8 N (coating thickness of 50 μm) and an 88.5% elastic recovery value [40]. Jeong et al. reported that polysilsesquioxane with ECTMS was flexible even in a folding test repeated 100,000 times at a radius of curvature of 3 mm [41].

In this study, we synthesized POSS and LPSQ based on ECTMS that exhibited flexible hard coating characteristics. We also analyzed the effect of the silsesquioxane structure on flexible hard coating. In particular, the LPSQ was synthesized using only ECTMS and, through a reaction time control, showed a higher molecular weight than the reported ladder-type polymers [43]. The synthesized LPSQ displayed a high molecular weight and high flexibility due to the regular arrangement of the flexible Si–O–Si chain.

## 2. Materials and Methods

### 2.1. Materials

2-(3,4-Epoxycyclohexyl)ethyltrimethoxysilane (ECTMS, 97%) was purchased from Gelest (Philadelphia, PA, USA). Tetramethylammonium hydroxide (TMAH, 10 wt% in water) was purchased from TCI (Tokyo, Japan). Isopropyl alcohol (IPA, 99.9%), toluene, tetrahydrofuran (THF, 99.9%), chloroform (99.8%), and water were purchased from Samchun (Seoul, Korea). All organic solvents were HPLC grade. Potassium carbonate (PC, 99%) was purchased from Sigma-Aldrich (St. Louis, MO, USA). Irgacure 250 (photoinitiator) was purchased from Shinyoung Rad Chem (Seoul, Korea).

### 2.2. Synthesis of EPOSS and EPSQ

EPOSS was synthesized by a sol–gel reaction. First, TMAH (10 wt% in water) was diluted with water to make a 5% solution. A mixed solution of IPA (4 mL) and TMAH (5 wt% in water, 3.9 g, 0.002139 mol) was added to a two-necked round-bottom flask. A mixture of IPA and ECTMS (15.24 g, 0.06 mol) was added to this flask and stirred for 40 min at 30 °C. The reaction proceeded for 6 h at 30 °C. Then, the IPA and water were removed from the rotary evaporator. Toluene was added to this flask, and the reaction proceeded for 4 h at 108 °C. After the reaction, an extraction was performed three times with toluene and water. The organic layer was dried over MgSO_4_ and filtered. EPOSS was obtained by removing the toluene using a rotary evaporator [44].

EPSQ was synthesized by a sol–gel reaction. Potassium carbonate (0.03 g, 0.00021 mol), water (3.6 mL, 0.20 mol), and THF (13.5 mL) were added to a one-necked round-bottom flask and stirred for 30 min. Subsequently, ECTMS (14.31 mL, 0.06 mol) was added dropwise to this flask. The reaction proceeded for 5 days at room temperature. Then, the THF was removed using a rotary evaporator to obtain a high-viscosity product, which was then dissolved in chloroform. Extraction was performed three times with water. The organic layer was dried over MgSO_4_ and filtered. EPSQ was obtained by removing the chloroform using a rotary evaporator [39].

### 2.3. Fabrication of EPOSS and EPSQ Films

EPOSS and EPSQ were mixed with a photoinitiator (Irgacure250). The solutions were mixed in a Thinky mixer (ARE-310, Thinky, Laguna Hills, CA, USA) for 5 min at 2000 rpm and then defoamed for 1 min at 2000 rpm. The solutions were uniformly applied on a polyimide (PI) film of 80 μm with a bar coater (ComateTM 3000VH, Kipae, Seoul, Korea). The coated films were dried on a hot plate for 2 min at 80 °C. The dried films were cured with UV irradiation (2000 mJ) using a UV curing equipment (DMH-1200, DTX, Seoul, Korea). The thickness of the film coatings was 25 μm, which was measured using a micrometer (IP65 Digital Outside Micrometer, Mitutoyo, Japan).

### 2.4. Characterization of the EPOSS and EPSQ Films

The chemical structures of EPOSS and EPSQ were confirmed by ^1^H nuclear magnetic resonance spectrometry (^1^H NMR, Advance 300, Bruker, Germany) and ^29^Si NMR (AscendTM500, Bruker, Germany). NMR analyses of EPOSS and EPSQ were performed using solutions of CDCl_3_ with tetramethylsilane (TMS) as the standard. The chemical structures of EPOSS and EPSQ were confirmed by Fourier transform infrared spectroscopy (FT-IR, Nicolet 6700, Thermoscience, Waltham, MA, USA) using the attenuated total reflection (ATR) method. The molecular weights of EPOSS and EPSQ were measured using gel permeation chromatography (GPC P-4000, Futecs, Daejeon, Korea) with a refractive index (RI) detector. The thermal properties of EPOSS and ESPQ were obtained by thermogravimetric analysis (Pyris 1 TGA, PerkinElmer, Seoul, Korea) under nitrogen gas at a heating rate of 20 °C/min from 30 to 700 °C. T_d10_ is the decomposition temperature at 10% weight loss. The surface roughness of the EPOSS and EPSQ films was studied by atomic force microscopy (AFM) using a scanning probe microscope (SPM-9700, Shimadzu, Japan). The AFM was performed by tapping mode using cantilever (NCHR-50, 320 kHz). The scan size was 5 × 5 μm. The maximum roughness (Rz) and the average roughness (Ra) were analyzed by AFM. X-ray diffraction (XRD) was performed on an X-ray diffractometer (SmartLab, Rigaku, Japan) in the 2θ range from 5 to 35° using Cu Kα irradiation (λ = 1.54 Å). The optical properties of the films were measured using a haze meter (HZ-V3, SUGA Test Instruments, Tokyo, Japan) and ultraviolet–visible (UV–vis) spectrometer (Lambda 750, PerkinElmer, Shelton, CT, USA). UV–vis spectroscopy was performed in the wavelength range of 200 to 800 nm. The hardness of the films was analyzed using a pencil hardness test and nanoindentation (Nanoindenter NHT^3^, Anton Paar, Graz, Austria). The pencil hardness test was performed in 1 kg at an angle of 45°. The nanoindentation test was performed with a Vickers diamond at a maximum force of 250 mN. The nanoindentation was repeated 5 times, and the median value of hardness was used as data. Folding tests on the films were performed with slide glass (microscope slides thickness approximately 1 mm, Paul Marienfeld GmbH & Co. KG, Lauda-Königshofen, Germany), and a home-made folding tester. During this folding test, one slide glass was used in the in-folding test (radius of 0.5 mm), and 10 glass slides were used in the out-folding test (radius of 5 mm). The folding test was performed at a radius of 1.5 mm for 100,000 cycles. After the folding test, the films were examined using an optical microscope (Olympus BX51, Olympus, Tokyo, Japan). Theoretical analyses were performed using density functional theory (DFT), as approximated by the B3 LYP functional and employing the 6-31G^*^ basis set in Gaussian09.

## 3. Results

### 3.1. Preparation and Characterization of Silsesquioxanes

As shown in Scheme 1, EPOSS and EPSQ were synthesized through a sol–gel reaction using 2-(3,4-Epoxycyclohexyl)etyhlmethoxysilane (ECTMS) as a monomer. In the sol–gel reaction, hydrolysis and condensation of ECTMS proceeded. As mentioned previously, factors influencing hydrolysis and condensation include the solvent, temperature, water-to-alkoxide molar ratio, and presence of acid or base catalysts [37,38]. Here, we synthesized EPOSS and EPSQ by controlling the hydrolysis and condensation by introducing different base catalysts.

EPOSS was synthesized by introducing tetramethylammonium hydroxide (TMAH), which was synthesized by introducing potassium carbonate (PC). The methoxy group of ECTMS was hydrolyzed and condensed by TMAH. The structure synthesized under controlled conditions had a polyhedral oligomeric silsesquioxane (POSS) mixture structure [37]. In the EPSQ synthesis, the methoxy group of ECTMS was hydrolyzed and polycondensed by potassium carbonate to form a ladder-type structure with a high molecular weight. [32].

^1^H nuclear magnetic resonance (NMR) and ^29^Si NMR were performed to analyze the chemical structures of the synthesized EPOSS and EPSQ (Figure 1). As shown in Figure 1a, ECTMS used as a monomer exhibited a peak for the methoxy group at approximately 3.55 ppm and showed an overall sharp peak. In contrast, in EPOSS and EPSQ, this methoxy group peak was decreased due to the sol–gel reaction. This is because Si–O–Si bonding was formed by the hydrolysis and condensation of methoxy groups [39]. In addition, the silanol group at 5.0 ppm was not found due to full condensation. ECTMS had a sharp peak as a monomer, whereas EPOSS and EPSQ had broad peaks, such as those that occur in oligomers and polymers [7,45].

The detailed molecular structures of EPOSS and EPSQ were characterized using ^29^Si NMR spectroscopy. As shown in Figure 1b, ECTMS used as a monomer showed a sharp peak in the Si–OCH_3_ group at -42 ppm. On the other hand, in the synthesized EPOSS and EPSQ, various peaks appeared in the upfield compared to ECTMS due to the formation of Si–O–Si bonds. EPOSS showed various sharp T^3^ (alkyl-Si(OSi-)_3_) peaks in the range of -65 to -70 ppm, which corresponded to the T_6_, T_8_, T_10_, and T_12_ structures. Those peaks proved that EPOSS had a POSS mixture structure [46,47].

On the other hand, EPSQ showed a broad peak in the range of −65 to −70 ppm, mostly from the T^3^ structure. The ladder-type structure composed of the T^3^ structure showed a strong intensity and a broad peak in the ^29^Si NMR due to polycondensation [48,49]. As shown in Figure 1c, various types of silicone bonds existed. The ^29^Si NMR results showed that, among the various silicon bonds, EPOSS and EPSQ had a structure consisting of mostly of T^3^. The chemical structures of EPOSS and EPSQ were further characterized using Fourier transform infrared spectroscopy (FT-IR), as shown in Appendix A. In EPOSS, the Si–OCH_3_ peak decreased at 1080 cm^−1^, and a Si–O–Si peak occurred near 1100 cm^−1^. The FT-IR results confirmed that EPOSS was synthesized in a POSS mixture structure. The FT-IR spectra of the synthesized EPOSS were similar to those of known POSS mixtures [45,50]. On the other hand, EPSQ had two Si–O–Si bonding peaks due to the unique double-stranded structure of the ladder-type structure [39]. The Si–O–Si symmetric stretching vibration of EPSQ occurred at 1020–1060 cm^−1^, and asymmetric stretching vibration occurred at 1100–1150 cm^−1^ [7,33,48]. After the sol–gel reaction, the methoxy peak at 1080 cm^−1^ decreased, and two Si–O–Si chain peaks were observed at 1030 and 1100 cm^−1^. EPSQ was confirmed to be a ladder-type structure composed of a double-stranded structure. In addition, EPOSS and EPSQ showed cycloaliphatic epoxy peaks at approximately 885 cm^−1^. This means that the sol–gel reaction did not affect the cycloaliphatic epoxy [51].

The molecular weights of EPOSS and EPSQ were analyzed using gel permeation chromatography (GPC). As shown in Table 1, the weight average molecular weights (M_W_s) of EPOSS and EPSQ were 1700 and 16,900, respectively, with different M_W_s depending on the base catalyst. EPOSS showed M_W_s similar to those in other studies [52], and EPSQ showed a higher M_W_ in the homopolymer structure compared to that in other studies [43]. This is the result of controlling the catalyst and reaction time in the sol–gel reaction. Increasing the Si–O–Si chain length is expected to increase the flexibility of the coating films. Overall, NMR, FT-IR, and GPC analyses indicated that EPOSS is a POSS mixture structure with a molecular weight of 1700 and EPSQ is a ladder-type structure with a molecular weight of 16,900.

### 3.2. Thermal Properties of Silsesquioxanes

A thermogravimetric analysis (TGA) was performed to investigate the thermal properties of the synthesized EPOSS and EPSQ. Appendix A shows the TGA curves of ECTMS, EPOSS, and EPSQ. In the TGA curves, ECTMS showed a rapid mass loss above 100 °C due to the lack of Si–O–Si bonds. On the other hand, EPOSS and EPSQ showed a weight reduction 6.20 and 8.54 wt%, respectively, due to the thermal decomposition of residual monomers and silanol groups from 100 to 450 °C. Above 450 °C, EPOSS and EPSQ showed rapid mass reduction due to the thermal decomposition of the organic functional groups. Finally, the TGA curves confirmed that more than 35% of the silica compound and carbon remained at 700 °C [32]. EPOSS showed a T_d10_ (temperature at 10 wt% loss obtained by TGA) of 468 °C, and EPSQ showed a T_d10_ of 459 °C. The TGA indicated that EPOSS and EPSQ had higher thermal stabilities than that of general carbon-based polymers [53,54]. The high thermal stabilities of EPOSS and EPSQ are due to the high bond dissociation energy of Si–O–Si. In general, the bond dissociation energy of the Si–O bond is 460 kJ/mol, which is high compared to that of C–O (345 kJ/mol) and Si–C (318 kJ/mol) [16,55]. Owing to the high bond dissociation energy, EPOSS and EPSQ composed of Si–O–Si bonds have a high thermal stability.

### 3.3. Characterization of Silsesquioxane Films

To investigate the coating properties of the synthesized EPOSS and EPSQ, a coating film was prepared with a photoinitiator. The coating films were applied to a thickness of 25 μm on a 80 μm PI film using a bar coater. After coating, the film was treated by heat and UV curing. FT-IR was performed to confirm the epoxy conversion of EPOSS and EPSQ after UV curing. Appendix A shows the FT-IR spectra before and after the UV curing of EPOSS and EPSQ. As shown in Appendix A, a decrease in the intensity of the epoxy ring peak was observed at 885 cm^−1^. This indicates the opening of the epoxy ring by UV radiation [50]. In addition, as shown in Appendix A, a hydroxyl group peak appeared near 3400 cm^−1^ due to the ring opening of the epoxy ring [40]. EPSQ also showed spectra similar to those of EPOSS. Thus, the conversion of epoxy by UV radiation was confirmed. Atomic force microscopy (AFM) was used to investigate the surface according to structural differences between the prepared EPOSS and EPSQ films. As shown in Figure 2a,b, in the AFM image, the EPOSS film showed a nonuniform morphology image and a maximum roughness (Rz) of 25.10 nm. This result was due to the aggregation of EPOSS molecules with strong interactions between the molecules [56]. Such aggregations may act as a defect in the coating film [57]. On the other hand, as shown in Figure 2c, the EPSQ film exhibited a more uniform surface image than EPOSS. The EPSQ film had a Rz of 6.81 nm (Figure 2d) and showed a much more uniform surface morphology than the EPOSS film. In addition, the EPOSS film and the EPSQ film showed average roughness (Ra) of 4.53 and 1.09 nm, respectively. This is because the ladder-type EPSQ forms from a regular molecular chain; therefore, aggregations such as in EPOSS do not occur, and the molecular chains are effectively arranged. For this reason, the EPSQ film showed a uniform morphology compared to the EPOSS film.

### 3.4. Structural Properties of Silsesquioxane Films

X-ray diffraction (XRD) was performed to analyze the orientation of the EPOSS and EPSQ films. As shown in Figure 3, in the out-of-plane diffraction pattern of the EPOSS film, broad diffraction peaks appeared at approximately 7.1 and 17.5°. The broad diffraction peak at 7.1° indicates the chain-to-chain length (d-spacing 1) of an interdigitated molecule by the side chain of the EPOSS molecule. [58] The d-spacing (d1) of the EPOSS film was 1.24 nm (λ = 2dsinθ). On the other hand, the broad diffraction peak at 17.5° indicates the average thickness (d-spacing 2) of EPOSS molecules. [58] The d-spacing (d2) of the EPOSS film was 0.50 nm. Unlike the EPOSS film, in the out-of-plane diffraction peak of the EPSQ film, broad diffraction peaks appeared at approximately 6.3 and 18.0°. The broad diffraction peak at 6.3° indicates the chain-to-chain length (d-spacing 1) of an interdigitated PSQ backbone by the side chain of EPSQ [48], and the d-spacing (d1) of the EPSQ film was 1.40 nm. On the other hand, the broad diffraction peak at 18.0° indicates the average thickness (d-spacing 2) of ladder-like polysilsesquioxanes [48]. The d-spacing (d2) of the EPSQ film was 0.49 nm. The EPOSS film showed a shorter d-spacing (d1) than the EPSQ film, which is due to the strong interaction between the EPOSS molecules, as shown in the AFM image [59].

The measured XRD patterns were similar to the XRD patterns of the POSS mixture and LPSQ [58]. In addition, EPSQ d-spacing 2 showed a stronger intensity than the XRD data from other studies with similar structures [29]. This is because the synthesized polymer has a higher region regularity than the polymers reported in the literature. Therefore, it is expected to show improved physical properties, such as hardness and flexibility. As shown in Appendix A, we analyzed the molecular structure using DFT calculations (Gaussian09) [60]. The results of XRD showed the d-spacing (d2) of EPOSS and EPSQ were 0.50 and 0.49 nm, respectively, which was similar to the thickness (0.35–0.51 nm) obtained by density functional theory (DFT) calculations. Overall, considering the DFT calculation and XRD data, both EPOSS and EPSQ are expected to have formed more pentagonal shapes than square shapes.

### 3.5. Optical Properties of Silsesquioxane Films

The optical properties of the prepared EPOSS and EPSQ films were confirmed using a haze meter and UV–visible spectrophotometer (UV–vis). Figure 4 shows the haze meter results. The ECTMS film exhibited properties that were optically similar to those of the bare PI film. On the other hand, the EPOSS and EPSQ films had a haze value of 0.9 or less, which was an improvement over the optical properties of bare PI film with a haze value of 1.1. In addition, the EPOSS and EPSQ films had more than 90.0% total transmittance, which was higher than the 89.5% of the bare PI film. This is due to the low refractive index of the silsesquioxane structure, which has a refractive index of 1.55 or less [9,61]. The refractive index has a significant influence on the optical properties of the film [62]. In particular, the transmittance of the film tends to be improved when a material with a low refractive index is coated on a substrate with a low transmittance [63,64]. For this reason, silsesquioxane-based materials with a low refractive index are used to improve optical properties [61,65]. Appendix A shows the UV–vis spectra. In the UV–vis spectra, the EPOSS and EPSQ films showed a high transmittance of more than 90% at 550 nm. The high optical properties of the EPOSS- and EPSQ-coated films were confirmed using a haze meter and UV–vis spectroscopy. In addition, cycloaliphatic epoxy–siloxane hybrid materials are known to exhibit high light transmittance and high transmittance stability at high temperatures [66]. These excellent optical properties were confirmed by the EPOSS and EPSQ coatings, which are based on cycloaliphatic epoxy with good optical properties.

### 3.6. Mechanical Properties of Silsesquioxane Films

Pencil hardness tests and nanoindentation analyses were performed to investigate the mechanical properties of the EPOSS and EPSQ films. The coating thickness of the films was 25 μm. In the pencil hardness test, the pencil was tilted at an angle of 45° to scratch the specimen with a fixed load. As shown in Table 2, under a 1 kg load, the EPOSS film had a pencil hardness of 5H, and the EPSQ film had a pencil hardness of 8H. The EPSQ film has a high pencil hardness value because it has a very rigid and regular double-stranded structure. Unlike the single-siloxane chain, the rigid double-stranded structure shows high rigidity due to the hindrance of the internal rotation of the Si–O bonds in the main chain [34]. In addition, because EPSQ forms a double-stranded structure through high condensation, EPSQ exhibits higher hardness characteristics than EPOSS [9,32]. In addition, the mechanical properties of the EPOSS coating film are expected to decrease owing to the aggregation of EPOSS, as shown by AFM.

As shown in Figure 5, to progress with the high-precision analysis of the EPOSS and EPSQ films, a nanoindentation test was introduced to investigate the mechanical behavior of the coating film. In the nanoindentation test, the indenter tip was pressed into the specimen until a prespecified maximum load was reached. Then, the load was removed, and the mechanical properties of the specimen were analyzed [67]. In general, as the molecular weight increases, mechanical properties such as hardness tend to improve [68]. Therefore, owing to its higher molecular weight, the EPSQ film showed a significantly higher hardness than the EPOSS film. As shown in Figure 5a, the load–displacement curve shows that the slope of the EPSQ film is larger than that of the EPOSS film. In general, in the nanoindentation test, a sample with high hardness has a large slope because it requires a large force to increase the indentation depth [23]. In other words, the EPSQ film has a large slope because of its high hardness. The EPOSS and EPSQ films have E* (effective Young’s modulus) values of 4.83 and 5.15 GPa, respectively. Here, E* was calculated as (E* = E/(1 − ν^2^)) using Poisson’s ratio (ν) [40]. A lower E* has an effect on the increase in free volume [23]. In the case of the EPOSS film, the free volume of the coating film increased owing to the aggregation with nonuniform coating. The EPOSS film with a large free volume exhibited a higher hysteresis loop. For this reason, the POSS film exhibited a 74.4% elastic recovery. In contrast, the elastic recovery of the EPSQ film was 80.1%, which was 5.7% higher than that of the EPOSS film. Musil et al. suggested that an elastic recovery of over 60% indicates a flexible hard coating [69]. Both the EPOSS and EPSQ films had elastic recoveries of greater than 60%. The EPSQ film with a low free volume showed an improved elastic recovery as well as high hardness. Additionally, the EPOSS and EPSQ films had H/E* values of 0.075 and 0.094, respectively. Here, a low H/E* value indicates plasticity. As the plasticity increased, the deformation area increased when stress was applied to the specimen. This means that the EPOSS film with a low H/E* had more deformed areas under external stress. On the other hand, the EPSQ film with a high H/E* inhibited plasticity, and the deformation area was reduced under external stress [70]. As a result, the EPSQ film exhibited high resistance to deformation. Figure 5b shows a plot of H/E* and the elastic recovery. Here, the EPSQ film had a higher H/E* and elastic recovery than the EPOSS film. This can be explained by the EPSQ film having less deformation and a higher elastic recovery to external stresses than the EPOSS film. EPSQ films with excellent mechanical properties are expected to be suitable for flexible hard coatings. Bae’s group plotted the wear resistance of polymers, metals, and ceramics with a hardness–effective modulus curve map [40]. As shown in Figure 5c, EPOSS films exist in the plastic field; however, EPSQ films show higher hardness and E* characteristics than plastic [71]. As a result, EPSQ with very rigid double-stranded structures and high molecular weights have better mechanical properties than the EPOSS structures.

As shown in Figure 6, in-folding and out-folding tests were performed to examine the flexibility of the EPOSS and EPSQ films. In the in-folding test, the coated surface was folded inward, and in the out-folding test, the coated surface was folded outward. First, as shown in Figure 6a, the in-folding test of the EPOSS film was performed using a glass slide with a diameter of 1 mm (r = 0.5 mm). As shown in Figure 6b, the occurrence of cracks in the EPOSS film after the in-folding test was confirmed by optical microscopy. With the same method, the in-folding test of the EPSQ film was performed using a glass slide, as shown in Figure 6c. In the optical microscopic analysis, the EPSQ film did not crack after the in-folding test (Figure 6d). The excellent mechanical properties are the result of the regularity of the double-stranded structure and the molecular-level hybridization of EPSQ [7,9]. As shown in Figure 6e,g, out-folding tests (r = 5 mm) of the EPOSS and EPSQ films were performed. The out-folding test was performed using 10 glass slides (r = 5 mm). As shown in Figure 6f, the occurrence of cracks in the EPOSS film after the out-folding test was confirmed by optical microscopy. However, as shown in Figure 6h, the EPSQ film did not crack after the out-folding test. Unlike the EPOSS film, the EPSQ film did not crack in either the in-folding or the out-folding tests. The flexibility of the film was improved owing to the regular structure of the long Si–O–Si chain of EPSQ. In contrast, the flexibility of the EPOSS film was reduced due to nonuniform surface morphology resulting from the aggregation between the molecules, as shown in the AFM image. Thus, the EPOSS film cracked during the folding test. The flexible EPSQ film was then subjected to 100,000 cycles of in-folding tests using a folding machine (r = 1.5). As shown in Figure 6j, cracks did not occur when examined by optical microscopy after the tests. Through this extensive folding test, the flexibility and stability of the EPSQ film were confirmed. In conclusion, owing to the regular arrangement of Si–O–Si, which is a flexible chain, EPSQ showed excellent coating properties, high hardness, and high flexibility. For this reason, EPSQ is expected to be applicable to flexible displays.

## 4. Conclusions

In this study, we synthesized EPOSS and EPSQ by controlling the catalyst and reaction time in the sol–gel reaction. The molecular weight of EPOSS and EPSQ were 1700 and 16,900, respectively, with different molecular weights depending on the base catalyst. The high molecular weight of EPSQ is due to polycondensation by potassium carbonate. In AFM analysis, the EPSQ film showed uniform surface morphology due to regular double-stranded structure, but the EPOSS film showed nonuniform surface morphology due to strong aggregation of the EPOSS molecules. The aggregation of EPOSS was also confirmed from XRD analysis. In XRD curves, the EPOSS film and the EPSQ film showed broad diffraction peaks of 7.1 and 6.3°, respectively. Therefore, the d-spacing (d1) of EPOSS and EPSQ were 1.24 and 1.40 nm (λ = 2dsinθ), respectively. The EPOSS film showed shorter d-spacing (d1) than the EPSQ film due to strong aggregation. In addition, the EPOSS film and the EPSQ film showed broad diffraction peaks of 17.5 and 18.0°, respectively. The d-spacing (d2) of EPOSS and EPSQ was 0.50 and 0.49 nm, respectively. Considering XRD analysis and the DFT calculation, both EPOSS and EPSQ are expected to have formed more pentagonal shapes than square shapes. The EPSQ film showed high hardness (5.15 GPa) and high elastic recovery (80.1%) in the nanoindentation test due to regular double-stranded structure and uniform surface morphology. Moreover, the EPSQ film showed high flexibility without crack in the in-folding (r = 0.5 mm) and out-folding (r = 5 mm) tests. On the other hand, due to aggregation, the EPOSS film showed a lower nanoindentation hardness and was more easily cracked than the EPSQ film.

## Data Availability

The data presented in this study are available on request from the corresponding author.

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
