# Peer review of "Hard Coating Materials Based on Photo-Reactive Silsesquioxane for Flexible Application: Improvement of Flexible and Hardness Properties by High Molecular Weight"

_polymers, 2021, doi:10.3390/polym13101564_

Round 1

Reviewer 1 Report

The job is interesting. Some aspects require clarification:
1. What operating mode and operating conditions / cantilever of the AFM microscope have been used?

2. What roughness parameter was specified (at least two are popular: RMS and Ra)

3.If the AFM measurements were made in dynamic mode, do the authors have phase contrast images?

4. The descriptions of the AFM image axes are very small, I cannot see the line along which the cross-section is made.There is no scale information for the x, y axis. Guess it's micrometers. Is it true? There is no scale information for the x, y axis. Guess it's micrometers. Is it true (Figure 2)?

5. What is the error for transmittance measurements in Figure 4? Please put it on the chart bars.

6. What was the thickness of the sample used for the indentation.
Was one or more indentation test performed and what is the measurement statistics?

Author Response

We are happy to have good comments from the reviewer. The manuscript has been revised based on the comments, point by point.

Reviewer 2 Report

Two epoxy functionalized silsesquioxanes with different topologies, i.e. ladder- and cage-type,  have been successfully prepared by using different bases in this paper. They are further used to prepare films by photopolymerization. The authors provide detailed data on synthesis and characterization and explain the reason the difference between two types of silsesquioxane. Some review on silsesquioxanes-related materials should be cited in this paper to attract more people to care about this field, for example,Dalton Trans.,2020, 49, 5396–5405.  

Author Response

(The authors gave the same response as above.)
